# Yeast Chronological Lifespan: Longevity Regulatory Genes and Mechanisms

**DOI:** 10.3390/cells11101714

**Published:** 2022-05-23

**Authors:** Mario G. Mirisola, Valter D. Longo

**Affiliations:** 1Department of Surgery, Oncology and Oral Sciences, University of Palermo, Via del Vespro 129, 90127 Palermo, Italy; 2Department of Biological Sciences, Longevity Institute, Leonard Davis School of Gerontology, University of Southern California, Los Angeles, CA 90089, USA; 3IFOM, FIRC Institute of Molecular Oncology, 20139 Milan, Italy

**Keywords:** chronological lifespan, aging, yeast longevity, bioactive substances, pro-longevity factors

## Abstract

*S. cerevisiae* plays a pivotal role as a model system in understanding the biochemistry and molecular biology of mammals including humans. A considerable portion of our knowledge on the genes and pathways involved in cellular growth, resistance to toxic agents, and death has in fact been generated using this model organism. The yeast chronological lifespan (CLS) is a paradigm to study age-dependent damage and longevity. In combination with powerful genetic screening and high throughput technologies, the CLS has allowed the identification of longevity genes and pathways but has also introduced a unicellular “test tube” model system to identify and study macromolecular and cellular damage leading to diseases. In addition, it has played an important role in studying the nutrients and dietary regimens capable of affecting stress resistance and longevity and allowing the characterization of aging regulatory networks. The parallel description of the pro-aging roles of homologs of RAS, S6 kinase, adenylate cyclase, and Tor in yeast and in higher eukaryotes in *S. cerevisiae* chronological survival studies is valuable to understand human aging and disease. Here we review work on the *S. cerevisiae* chronological lifespan with a focus on the genes regulating age-dependent macromolecular damage and longevity extension.

## 1. Introduction

The budding yeast *Saccharomyces cerevisiae*, a fungus commonly employed for brewing, wine and bread making since the time of Ancient Egypt, is also a simple and potent eukaryotic model system. The reasons for its success reside in the rigorous molecular genetic techniques used, the high throughput methodologies, the bioinformatic resources, and its low cost and amenability to genetic and biochemical techniques, in terms of which it is second only to the prokaryote Escherichia coli. Even before the era of systematic sequencing projects began it was clear there were several orthologous genes in yeast and mammals. One of the first examples was the ability of human Ras to reverse the phenotype of ras1ras2 double gene deletions in yeast [1].

When the entire genome was sequenced in 1996 many additional yeast genes were found to have a mammalian counterpart and the eventual availability of the human genome sequences confirmed the existence of a yeast orthologue for at least 30% of disease-causing human genes [2,3]. 

These properties were used for the understanding of a wide range of basic cellular processes, as confirmed by the number of Nobel prizes granted for scientific discoveries related to yeast made in the last 20 years (2001 cell cycle, 2006 molecular basis of transcription, 2009 telomere and telomerase, 2013 vesicle trafficking and 2016 autophagy). In addition to the general advantages of yeast as a model system, its short lifetime, the availability of precise high throughput techniques, and finally the possibility of gathering billions of isogenic individuals at a very low cost established it as the organism of choice for many researchers. It is therefore no surprise that the *S. cerevisiae* chronological and replicative (based on budding potential) lifespans became important pillars in aging research. 

## 2. The Methods

The first attempt to measure the yeast lifespan can be backdated to Mortimer and Johnston in 1959 [4]. This lifespan was based on determining the budding potential of an individual mother cell. Instead of splitting the mother cell into two almost identical daughter cells, as other eukaryotes do, the *S. cerevisiae* daughter cell emerges as a growing bud clearly distinguishable from the mother cell. The emerging daughter cell can thus be removed by micromanipulation as soon as it becomes visible, and the process can be repeated until the mother cell loses the capability to divide, enters a post-replicative phase, and eventually dies. This first experiment, conducted using various yeast diploid cells, revealed that a mother cell can undergo a range of 9–43 cell divisions. The authors pointed to the chitin scars accumulated on the cell wall after each budding as a possible cause of aging, or alternatively suggested that the observed larger size of aging cells will change the surface to volume ratio and that this ratio cannot surpass a certain limit. This second hypothesis was supported by several studies both in yeast and in mammalian cells [5,6].

Curiously, no further usage of this straightforward method, called replicative lifespan (RLS), is recorded until the 1990s, when this method, slightly improved, was combined with newer genetic techniques to identify genes and pathways involved in the aging process. 

The established protocol for RLS determination includes microdissection of daughter cells from mother cells, taking advantage of a microscope equipped with a micromanipulator developed for tetrad dissection. The detailed protocol is fully described elsewhere and is not the focus of this review [7]. However, the procedure involves obtaining single cells from the frozen stocks, creating a coordinated system within each plate to precisely position daughter cells, avoiding any possible confusion, and isolating single virgin daughter cells. At a 160× magnification the different size of mother and daughter cells easily allows the unequivocal recognition of each. Due to the time-consuming procedure, alternative methods have been proposed. One of the most recent is electrical impedance microscopy, an alternative noninvasive and much less time-consuming method to perform replicative lifespan measurements [8].

More recently, a different paradigm to measure yeast lifespan has been developed called chronological lifespan (CLS) [9,10]. Yeast grown on liquid media divides up to a certain cell density and then enters the post-diauxic phase, characterized by a switch to elevated use of non-fermentable carbon sources and high respiratory rates. Cultures kept in this condition maintain viability for a period of time ranging from just a few days to a week, depending on the genetic background. The number of living cells is determined by measuring their capability to generate new colonies when shifted to nutrient-rich media. Therefore, this method measures how long yeast cells can survive and remain functional in a non-dividing state phase. Temperature, amount and quality of available nutrients, pH, acetic acid concentration, and genotype are known to affect the survival time [11]. It is worth noting that cultures in synthetic dextrose media reach the highest cell density after 2–3 days, but their metabolic state remains high for an additional 4–6 days [12]. Therefore, the idea that CLS mimics a condition of slower metabolism or of starvation is incorrect, as confirmed by the discovery that when cell cultures start to die, glycogen content and extracellular ethanol as well as other nutrients remain available [13]. However, eventually, the surviving cell population enters a lower metabolic rate termed the stationary phase, during which survival can continue. 

Among the characteristics of aging yeast cells are apoptotic markers [14,15], involving a process that may have been advantageous for populations of single cell organisms facing great fluctuations in nutrient availability including carbon and nitrogen sources. In fact, apoptosis may be part of an “altruistic aging” program leading to the generation of better adapted mutants that take advantage of the nutrients released by the dead cells to survive and grow [14].

The basic CLS protocol is performed in liquid cultures, but it starts by streaking yeast cells from a frozen stock on nutrient rich dextrose-based agar plates (YPD). After 2–3 days, depending on the genotype, single colonies are transferred to flasks containing liquid synthetic media with a liquid–air ratio of 1:4 that grow at 30 °C with vigorous shaking. Small aliquots are collected every 1–2 days, serially diluted, and plated on YPD plates to obtain countable colonies, namely the colony forming units (CFU). This number increases for the first days of incubation, reaching the maximum concentration after 2–3 days of incubation, then levels off and eventually decreases. The number of colonies obtained at day three is thus considered the 100% survival level and is used as a reference to quantify the survival at the other time points. The latter is thus expressed as a percentage of the colonies measured at day three 

The way in which chronological lifespan is measured has undergone several variations over time to improve accuracy and automation. High throughput or semiautomated processes were developed and differences due to the way the CLS was performed appeared. The original flask-based method has thus been paralleled by growth on microplates, microfluidic methods and measurement of lifespan in agar-based plates. Interestingly, the latter method found clear differences between the cells at the top of the colonies and the cells at the bottom, also suggesting a crosstalk between cells of different parts of the colony [16]. It has also been argued that part of the cells that have lost the capability to reenter the cell cycle and are thus unable form colonies could still be alive and that this should be considered when measuring CLS since an underestimate of the live–dead cell ratio is possible. Dyes capable of selectively staining only live/dead cells have then been used to discriminate dead cells from cells that have simply lost the ability to reenter the cell cycle, and it has been found that cell death correlates with loss of colony-forming potential [17].

A role for acetic acid as a toxic factor limiting survival has been proposed by Burtner [18] and supported by the effect of a mutation in the Ach1 gene, which is responsible for acetyl coenzyme A production and whose deletion provokes acetic acid accumulation [19]. However, we and others [20] have suggested, based on several observations, that the toxic effect of acetic acid is due to its role as a pro-aging carbon source similar to glucose or ethanol and not as a toxic extrinsic factor. Protein oxidation is acetic acid-dependent and not simply pH-dependent [21]. It has also been observed that a higher acetic acid concentration increases the accumulation of storage molecules such as glycogen and trehalose, thus confirming the role of acetic acid as a starvation-associated carbon source analogous to ketone bodies in mammals [21]. However, in yeast, a higher glucose concentration triggers acetic acid production which in turn stimulates mitochondria fragmentation, a phenotype independent of acidification [22]. Finally buffering of the media during CLS does not increase longevity, thus confirming that the role of acetic acid is not simply related to media acidification [23]. 

## 3. Mutation Rate Detection during CLS 

Spontaneous inactivation of the arginine permease CAN1 can be used to monitor the appearance of genetic mutations causing this inactivation [24]. Yeast uses this protein to uptake arginine as well as its toxic counterpart canavanine. Incubation of yeast in the presence of 60 mgL^−1^ of this substance inhibits growth unless a mutation impairs the capability of CAN1 to uptake canavanine. Arginine and canavanine compete for the same transporter and thus arginine must be absent from the media, limiting the assay to yeast able to biosynthesize it [25]. The experiment is performed as a standard CLS but, whenever an aliquot of the culture is diluted to count the CFUs, an aliquot of 2 × 10^7^ cells is washed with water and incubated on plates lacking arginine but containing canavanine. The mutation frequency is calculated as the number of colonies appearing in the presence of canavanine with respect to the total number of viable cells at that timepoint. Sequencing of the CAN1 gene during CLS found mainly nucleotide substitutions (65%), followed by deletions or insertions (25%) and a minority (10%) of more complex events [24]. To discriminate between these possible molecular events, the rate of nucleotide substitution can be precisely identified by measuring the reversion rate of a premature stop codon. The widely used DBY746 strain has a C-T transition at codon 403 which generates a premature amber stop codon within the coding sequence of the TRP1 gene. Yeast cells carrying this mutation need Trp supplementation in the media to grow unless a point mutation reverts the amber codon to a coding codon [26]. Since the reversion at a specific locus is a quite rare event, the number of cells that must be plated to perform this assay is at least 10-fold higher than the number used to identify CAN revertants. Identification of insertion or deletion mutations can instead take advantage of the lysine prototrophy of an engineered yeast strain. In this case the Lys2 gene is engineered through enzyme restriction with *Bgl*II followed by Klenow treatment. The fragment reintroduced into the yeast genome generates a frameshift making yeast lysine auxotroph. Only insertion or deletion mutations restoring the correct reading frame of the Lys2 gene will allow yeast to grow in the absence of lysine.

Large chromosomal rearrangements are detected by replacing one of the members of the hexose transporter (HTX13) with the wild type URA3 gene in an ura3-genetic background. Aliquots of 100 million cells are then plated with the contemporary presence of canavanine and fluorotic acid. As the inserted URA3 and CAN1 are relatively close within the same chromosome, and since only cells which have lost both CAN1 and URA3 can form colonies, this method makes it possible to estimate the frequency of a chromosomal rearrangement [27].

## 4. Genetic Determinants of Longevity Discovered by Chronological Lifespan

### 4.1. Ras/PKA Pathway

The effect of the deletion of the gene coding for the Ras2 protein on longevity extension and stress resistance allowed the identification of Ras2 as one of the very first longevity regulatory genes in any organism [28]. RAS2 deletion doubles CLS and increases the resistance to heat and oxidative stresses [29,30] (Figure 1). Ras2 codes for one of the two monomeric G-proteins that are capable, when GTP-bound, of stimulating adenylate cyclase, which in turn activates PKA. The involvement of PKA as a pro-aging pathway was further confirmed by the observation that mutants with impaired synthesis of cAMP were also long-lived [29]. The Ras pathway appears to have a pro-aging role conserved from yeast to mammals, since in mice maternal uniparental disomy of the Ras exchange factor coding gene RasGRF (which is inactivated during oogenesis) has been associated with increased longevity [30], a hypothesis confirmed by the lifespan extension in mice carrying homozygous deletion of the gene coding for this nucleotide exchange factor [31]. Even though one could argue that Ras is not the only target of the exchange factor Ras-GRF, thus challenging the involvement of the PKA pathway, the observation that mice carrying deletion of AC5, one of the adenylate cyclase subunits, live longer further supports the role of PKA as a conserved pro-aging pathway. It has long been known that starved yeast shows a cAMP spike dependent on the CDC25-Ras2 pathway when glucose is added to the culture [32]. Even though glucose restriction also increases stress resistance in ras2-deleted cells [33,34], Ras-PKA is known to inhibit stress resistance transcription factors Msn2–4, thus reducing cellular protection [34,35,36] (Figure 2). Recent observations also suggest a link between glucose metabolism and epigenetic changes [37]. Depletion of carbohydrate storage through disruption of HAD, a class II HDAC, results in increased h3k18 acetylation, increased transcription of storage carbohydrate genes and longer lifespan [38].

In addition, deletion of the TDH2 gene that encodes for glyceraldehyde 3-phosphodeydrogenase, an essential enzyme for glucose metabolism (both for glycolysis and gluconeogenesis), is capable of suppressing the sensitivity to DNA damage obtained through HDAC impairment. Suppression of intrachromosomal recombination as well as an effect on replicative lifespan can be observed, suggesting that Ras signaling is at the crossroads between glucose metabolism, DNA stability, and longevity [39]. 

### 4.2. TOR/Sch9

The second major pathway capable of controlling aging in yeast is TOR, a serine-threonine protein kinase acting upstream of Sch9 [30,40]. The role of this pathway in regulating aging has been since confirmed in worms and flies [41,42]. Inhibition of Tor-S6k activity increases both chronological and replicative lifespan [29,35,43] (Figure 1). Decreased Tor signaling increases respiration [44] as well as mitochondrial protein expression [45], and reactive oxygen species are also increased, likely activating an adaptive response that extends CLS. Manipulation of yeast genome suggests Tor and Sch9 (the yeast orthologue of mammalian S6 kinase) act on the same pathway in response to amino acid signaling (Figure 1 and Figure 2), with serine, threonine and valine having a critical role. A link between nutrient availability, telomere length, the Tor protein kinase and CLS has also been proposed [46].

Evolutionary conservation is also supported by the observation that Tor-S6K signaling inhibition by rapamycin treatment increases the lifespan of mice [47] and genetic impairment of mTOR extends survival in a progeria mouse model [48]. Interestingly both Sch9 and S6 kinase are nutrient-dependent translational regulators [49]. TOR is also involved in general transcriptional regulation and, in fact, one of its targets, Maf1, which inhibits the activity of DNA Pol III in the nutrient-deprivation condition, increases the lifespan of yeast, worms, and flies and has a critical role in transcriptional regulation. The latter is confirmed by the observation that Maf1 deletion increases DNA Pol III concentration and shortens lifespan [50]. The role of transcriptional derangement during aging is confirmed by single-cell gene expression analysis, showing an increased intercellular gene expression heterogeneity during the aging process [51]. In addition, mutation in subunits of the SAGA complex (Spt-Ada-Gcn5 acetyltransferase; e.g., DUB, SPT7), a yeast–human conserved multiprotein complex that acetylates and deubiquitinates both histone and non-histone proteins, elicits both extended chronological and replicative lifespans [52].

### 4.3. Genomic Instability

Genomic stability is an important factor in delaying aging and diseases in mice and humans [53,54], and many genes regulating genome stability affect yeast longevity as well [55]. This idea was further strengthened by the observation that overexpression of inhibitors of pro-aging genes such as Mig1 and PDE2 are capable of addressing the longevity impairment observed in anaphase, promoting a complex deficient strain [56]. Higher order chromatin structures affect CLS and protect from UVB stress [57]. A systematic analysis of different kinds of mutations confirmed that genome instability due to recombination rates directly parallels longevity [58]. Critically shorter telomeres are associated with reduced lifespan, although increased length of telomere is also linked to a shorter lifespan. This apparent contradictory observation has been explained with regard to the re-localization of Sir gene family products from telomere to non-telomere sites. It is thus possible that the opposite role of Sir2 in CLS and RLS, whose deletion shortens the former and extends the latter, likely reflects the different roles of telomere homeostasis in dividing and non-dividing cells [13,59]. This observation, first made in RLS, is paralleled by the observation that doubling the length of telomeres does not increase yeast CLS [60]. The different behavior of cells in the growth or post-diauxic phases regarding genome stability has recently been confirmed by the analysis of DNA repeat instability mechanisms in a yeast model of Friedreich’s ataxia, where dividing cells undergo errors of DNA synthesis during repair or replication while non-dividing cells undergo larger deletions, with a median size of 500 bp, deriving from non-homologous end joining recombination (NHEJ), with a critical role played by MutSbeta, which is also involved in gene conversion [61]. Interestingly, mutations in the ubiquitin ligase BUL2, involved in amino acid uptake and telomere length homeostasis, increase CLS, suggesting a possible link between amino acid availability, telomere maintenance, and longevity [62]. Substances capable of specifically affecting genome stability in yeast have been discovered; for example, coffee has been found to increase CLS, possibly by protecting cells from reactive oxidative species and double DNA strand breaks [63], and astaxanthin has been demonstrated to reduce the mutation rate in DNA repair deficient mutant yeast (rad1∆, rad51∆, apn1∆, apn2∆, and ogg1∆), also increasing the CLS [64]. 

### 4.4. Amino Acid Availability

The idea that an isocaloric diet with different dietary compositions may differentially affect longevity was first proposed with regard to mice [65,66] and later studied in Drosophila melanogaster [67] and organisms ranging from yeast to humans [68,69]. Protein and amino acid restrictions, but also increases, can have a role in delaying aging through molecular mechanisms that have at least in part been identified [70,71] (Figure 1). Low methionine and high glutamic acid independently increase lifespan in an acetic acid-independent fashion [72]. Proline supplementation, which is known for its role not only as a building block but also as an energy source and stress protectant, is capable of increasing CLS in a proline oxidase (PUT1)-dependent mechanism [73]. Serine depletion has been demonstrated to increase lifespan [49]; however, other researchers comparing the amino acid content of calorie-restricted vs standard cultures identified serine and asparagine as enriched in calorie-restricted exhausted media and speculated about their possible pro-longevity effect. However, only when added at a non-standard concentration (sixfold) was serine was confirmed as a pro-longevity nutrient and, curiously, no dose-dependent effect was observed [74]. The discrepancy between these two observations, however, could also be linked to different amino acid compositions/concentrations and to the different genetic backgrounds used. The lack of essential amino acids has also been demonstrated to increase chronological lifespan [75], and valine and threonine restriction may contribute to CR-dependent CLS extension by down-regulating the TOR-Sch9 pathway through the phosphorylation of Sch9 [49] (Figure 2).

### 4.5. Inhibition of Ras-PKA Signaling

Due to its role as a critical oncogene in many cancers, there have been extensive efforts in the past decades to identify drugs with the ability to impair Ras activity. It must be noted that while direct targeting of the Ras proteins has remained elusive, leading to the classification of Ras as undruggable, its strict dependence on glycolysis provides a rationale for targeting glucose metabolism as a feasible alternative. Glucose restriction is a key regulator of Ras/PKA pathway activity in yeast and mammals, possibly through fructose 1,6 biphosphate, an intermediate product of glycolysis, which activates CDC25, the yeast orthologue of the mammalian SOS, stimulating the nucleotide exchange of Ras proteins [76] (Figure 1). In addition, vitamin C supplementation was found to be effective in inhibiting glyceraldehyde 3 phosphate dehydrogenase, delaying proliferation of KRAS or BRAF mutant colorectal cancer cells [77]. Various natural or artificial products have been identified, such as the flavonoid silybin capable of inhibiting glucose transporters GLUT1–4 [78].

Inhibition of cAMP production, downstream of Ras, was achieved in yeasts with the drug triclabendazole. As expected, inhibition of the Ras-cAMP signaling was associated with nuclear translocation and activation of the stress resistance transcription factors Msn2–4, required for maximum longevity extension [79]. Stillbenes such as resveratrol, oxyresveratrol, and picetannol were also found to increase phosphodiesterase (PDE2; Table 1 and Figure 2) expression, which is known to inactivate Ras signaling through linearization of cAMP, resulting in an 18% longer chronological lifespan, and interestingly they were also capable of activating the mammalian counterpart PDE4 [80].

### 4.6. Inhibition of TOR-Sch9 Signaling

TOR-Sch9 is a major nutrient-dependent pro-aging pathway and nutrient deprivation, especially that of certain amino acids, can inhibit TOR activity and increase survival [49,81] (see also Section 4.4 on amino acid availability). However, many natural substances have the same ability. For example, the salvia extract cryptotanshinone has been reported to double CLS but only in the presence of wild type Tor1, Sch9, and Gcn2 alleles, suggesting that it inhibits the TOR-Sch9 pathway [82] (Table 1). In addition, the requirement for wild type GCN2, whose function is to phosphorylate eIF2 in response to the presence of uncharged tRNA, suggests that this substance phenocopies the effect of amino acid restriction on TOR-Sch9 signaling. A role of caffeine as a Tor inhibitor has also been suggested in fission yeast [83].

### 4.7. Substances Capable of Affecting Autophagy

Long-term viability of non-mitotic cells can also be dependent on cellular processes capable of degrading old components and replacing them with new ones. In this scenario amino acid homeostasis based on amino acid uptake and amino acid recycling, mainly via autophagy of mitochondria and other cellular components, is a process that contributes to the maintenance of cellular homeostasis by removing damaged structures. Genome-wide screening confirms autophagy, vacuolar protein sorting, and tRNA methylation genes as regulators of CLS [84]. Reversible tRNA methylation affects translation efficiency and is dependent on glucose availability [85], while aminoacylation of tRNA affects translation quality control and the TOR pathway [86], thus rendering posttranscriptional modification of tRNA a potential drug target. The role of the vacuole in yeast aging is also confirmed by shortened CLS in the pep4 mutant, which is counterbalanced by quercetin treatment [87]. Spermidine is an epigenome modulator in many living organisms that can act through deacetylation of histone H3 and inhibition of histone acetyl transferase, resulting in increased expression of autophagy-related genes [88]. Lifespan shortening was observed for green tea extract and berberine whilst the strongest pro-longevity effect (RLS) has been observed with pterocarpum marsupium [89]. Geniposide derivatives extracted from gardenia jasminoides ellis increased both yeast replicative and chronological lifespans. Gentiopicroside extracted from Gentiana was demonstrated to prolong both replicative and chronological lifespans in an ATG32-dependent fashion, suggesting the involvement of mitophagy [90] (Figure 2). The role of mitochondrial autophagy in yeast aging is supported by the effect of the mitochondria-associated degradation pathway (MAD) on chronological lifespan [91]. As the atg32 and atg2 genetic background shows increased expression of SOD genes, autophagy’s beneficial effects may involve both breakdown and protective processes. In fact, the benzoquinone-type ehretiquinone from the herbal medicine Onosma bracteatum Wall increases both replicative and chronological lifespans, likely through antioxidant and autophagy means since sod1,2 or atg2,32 gene deletions prevent its ability to increase longevity, thus reinforcing the link between antioxidants and autophagy pathways [92]. Autophagy-dependent vacuolar acidification is also essential for the effect of methionine restriction [93] and methionine auxotrophy in increasing yeast lifespan [94]. In addition, phosphatidiletanolammine stimulates autophagy and increases the CLS [95], and citrus flavonoids are effective in regulating, in a dose-dependent fashion, yeast CLS [96]. A possible link between amino acid homeostasis and autophagy has been proposed with a critical role for branched-chain amino acids and the transcriptional regulator of general amino acid control GCN4 [97]. Rapamycin supplementation, which is known to also stimulate autophagy in a wide range of organisms, is also well-established to increase yeast longevity, strengthening the link between autophagy and lifespan [98]. 

Experiments in fission yeast also point to the role of proteasome and autophagy in maintaining mitochondrial function in quiescent fission yeast [99]. Autophagy-dependent and independent degradation of macromolecules have been confirmed in the budding yeast [100]. Vms1/Cdc48, components of the ubiquitin/proteasome complex, are involved in mitochondrial protein degradation, a metabolic pathway necessary to maintain cellular viability [101]. Upregulation of a component (Blm10) of the proteasome machinery is capable of prolonging RLS, likely by enhancing autophagy [102]. Ehretiquinone also shows increased chronological lifespan through autophagy and Sir2 regulation [103]. In summary, there are many natural substances able to increase yeast lifespan, and they are likely to involve activation of autophagy and mitophagy to delay aging (see Table 1 for a list).

### 4.8. Oxidants Regulation 

The role of oxidative stress in aging and disease was first proposed by Harman [104]. According to this theory, molecular damages leading to aging are triggered by the highly reactive oxygen species produced mainly by the mitochondrial electron transport chain. This theory is supported by the observation that long lived mutants, isolated in different living organisms, display reduced levels of ROS production and increased levels of antioxidants enzymes [9,10]. In addition, overexpression of SOD or the Bcl-2 gene delays aging. Considering the role of the latter as an antiapoptotic factor, it is possible that aging cells undergo programmed cell death [14]. Apoptosis has been observed when treating yeast cells with hydrogen peroxide or acetate [105]. It has also been demonstrated that inactivating mutations on the two key pro-aging pathways (Ras2 and Sch9) result in a threefold higher level of mitochondrial Sod2 activity [14]. However, overexpression of single or multiple antioxidant-coding genes results in a maximum of 30% increased CLS. Thus, this effect is not comparable to the threefold extension observed from Sch9 deletion (Figure 1), suggesting that nutrient response signal transduction “master regulators”, such as Tor and Sch9/S6K, are much more potent in affecting aging by regulating many processes and not only antioxidants [29]. It must also be noted that an increased level of superoxide, for example obtained by Sod1 deletion, increases adaptive regrowth. The latter is a phenomenon observed when the viability of aging cultures reaches 1% of the maximum value and the remaining cells start growing again using the metabolites released by the fraction of dead cells (a phenomenon often misclassified as extended lifespan). However, while increased levels of superoxide stimulate the regrowth, the residual viability of the 1% culture is not extended with respect to wild type cells, suggesting a rationale for the evolutionary role of Sod genes [14].

More recent results confirm that reactive oxygen species have a more complex role during aging, encompassing both damaging and signaling effects [106], and suggesting also that transiently elevated mtROS may have a hormetic effect, ultimately leading to increased longevity. This postulates the existence of mitochondria-to-nucleus stress signaling pathways capable of affecting the epigenome and consequently regulating the expression of many genes. Several natural products individually or in combination were screened for their ability to affect growth rate, TOR inhibition, and extension or shortening of lifespan. Increased expression of SOD genes is pro-longevity, but the effect was lost in genetic backgrounds with null alleles of atg32 and atg2, stressing the role of mitochondria and of autophagy, although these results do not show that autophagy genes are required for the effects of antioxidant genes on longevity but may simply indicate that autophagy genes are required for a long lifespan. Substances like almond skin extract and chlorogenic acid increase SOD1,2 and Sirt1/Sir2 gene expression with a consequent increase in CLS [107]. Yeast cells deficient in antioxidant production have been used to screen for molecules capable of reverting the defects of these mutants. Astaxantin, magnolol, and the pterocarpan glyceofillin have been identified as protective molecules [108,109] (Table 1). However, the role of these substances as pro-survival or pro-aging factors seems to depend on the concentration used, as expected from a hormetic effect [110] (Table 1). In fact, the role of a low concentration of stressors in triggering a protective hormetic response is supported by a number of studies [111]. Treatment with the polyphenol compounds pyrogallol and myricetin also increased stress resistance and caused lower levels of intracellular oxidation and protein carbonylation in yeast, and myricetin increased CLS in the absence of sod2 [112]. In addition, cocoa-derived polyphenol-rich extract increases chronological lifespan [113] (Table 1). Cucurbitacin B, chemically classified as triterpene and widely distributed in cucurbitacee, brassicacee, and allicin, a thioester of sulfenic acid extracted from garlic, affect oxidative stress and CLS, the latter upregulating the central oxidative stress regulator Yap1 [114,115] (Table 1).

### 4.9. Lipid Metabolism

Sphyngolipid (ceramide, sphingosine, and sphingosine-1-phosphate) metabolites are actively involved in signal transduction pathways modulating several processes, including stress responses, aging, and apoptosis. They affect oxidative stress and CLS [116], and genetic manipulation of lipid synthesis increases stress sensitivity and shortens the lifespan [117]. The connection between lipid and glucose metabolism is supported by the sphingolipid ceramide which also has a role as modulator of hexokinase function and can thus be considered at the crossroads between glucose and lipid metabolism [118]. A link between sphyngolipid and mitochondrial function has also been proposed [119]. Treatment of yeast cells with PUFA increased ROS production, negatively affecting respiration and shortening lifespan [120]. A role for lipid metabolism has also been suggested by the pro-longevity effects of litocholic acid (LCA). LCA triggers age-dependent remodeling of mitochondria morphology, size, and number, thus suggesting a link [121] between mitochondria lipidome and mitochondrial redox biology [122], and also involving remodeling of glycerophospholipid metabolism. Other substances capable of affecting lipid metabolism have been found associated with increased lifespan, for example P21 extract from salix alba is capable of increasing CLS and reshaping lipid metabolism [123]. These data suggest an intimate connection between lipid metabolism/supplementation and cellular survival but also a strict interconnection between lipid and glucose metabolism as a modulator of CLS.

**Table 1 cells-11-01714-t001:** Natural substances affecting yeast longevity and mechanisms involved.

Affected Pathway	Substance	Effect on CLS	Note	Reference
Ras-PKA	Resveratrol	↑		[80]
	Oxyresveratrol	↑		[80]
	Picetannol	↑		[80]
TOR/Sch9	Cryptotanshinone	↑		[82]
	Caffeine	↑	Only fission yeast	[83]
Autophagy	Spermidine	↑		[88]
	Green tea	↓		[89]
	Berberine	↓		[89]
	Pterocarpum marsupium	↑		[89]
	Gentiopicroside	↑		[90]
	Ehretiquinone	↑		[92,103]
	Phosphatidiletanolammine	↑		[95]
	Citrus flavonoids	↑		[96]
Oxidant regulation	Astaxantin	↑↓	Dose-dependent	[108,109,110]
	Magnolol	↑↓	Dose-dependent	[108,109,110]
	Glyceofillin	↑↓	Dose-dependent	[108,109,110]
	Pyrogallol	↑		[112]
	Myricetin	↑	sod2 delta background	[112]
	Cocoa polyphenol	↑		[113]
	Cucurbitacin B	↑		[114]
	Allicin	↑		[105]
Lipid metabolism	PUFA	↓		[120]
	Litocholic acid	↑		[121,122]
	Salix alba extract	↑		[123]

↑ indicates increased or ↓ reduced lifespan; ↑↓ indicates increased or reduced lifespan depending on the concentration used.

## 5. Conclusions

Biological and medical research has widely taken advantage of the usage of model systems. The more limited ethical issues, the lower cost, and the package of bioinformatic, genetic, and molecular tools justify their usage in most research fields. In addition, aging research has taken advantage of the shorter lifespan of model systems to monitor the lifespan of several generations in a reasonable time. The yeast model system shares all these characteristics and adds the availability of billions of isogenic individuals, thus allowing precise discrimination between environmental and genetic causes of the phenotypes observed. From the discovery of Ras as a critical pro-aging pathway, yeast has been used to identify genes and pathways critical for cellular longevity, leading to the identification of the Sch9/TOR pathway as the second major pro-longevity pathway (Figure 2). The dependence of these pathways on nutrient availability triggered the identification of the differential roles of single nutrients on these pathways. Glucose and amino acids were the first to be characterized and the discoveries made in yeast have also been confirmed in mammals. However, the effect of these substances is more complex than expected; glucose can in fact also affect the epigenome and some amino acids may act not only as an energy source and as building blocks for the synthesis of endogenous proteins but also as signaling molecules. Many natural substances have been identified that are capable of increasing the lifespan of post-mitotic cells (CLS) and in some cases yeast has also allowed the identification of the cellular or molecular pathway affected by those substances in detail (See Table 1). For example, caffeine and salvia extract are capable of inhibiting TOR, the latter with a mechanism involving amino acid availability. Other screenings have identified the role of autophagy, oxidants, and lipids, as well as the substances capable of triggering or rescuing the strain-deficient phenotype of those pathways. The possibility of precisely discriminating between genetic and environmental causes of yeast phenotypes, as well as the possibility of screening for substances capable of rescuing those phenotypes. will keep yeast as a pillar of biomedical research.

## Figures and Tables

**Figure 1 cells-11-01714-f001:**
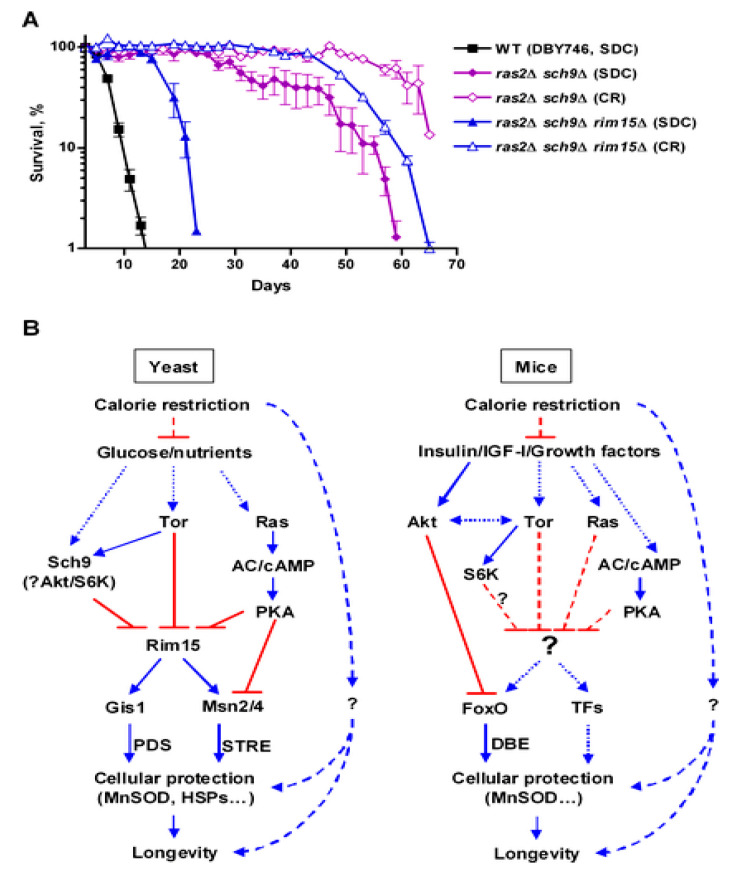
Relative contribution of genetic and nutrient manipulation on yeast lifespan. (**A**) CR cultures were switched to water-only at day 3 mimicking fasting condition. (**B**): Side by side comparison of yeast and mammalian aging pathways. Reproduced from [33].

**Figure 2 cells-11-01714-f002:**
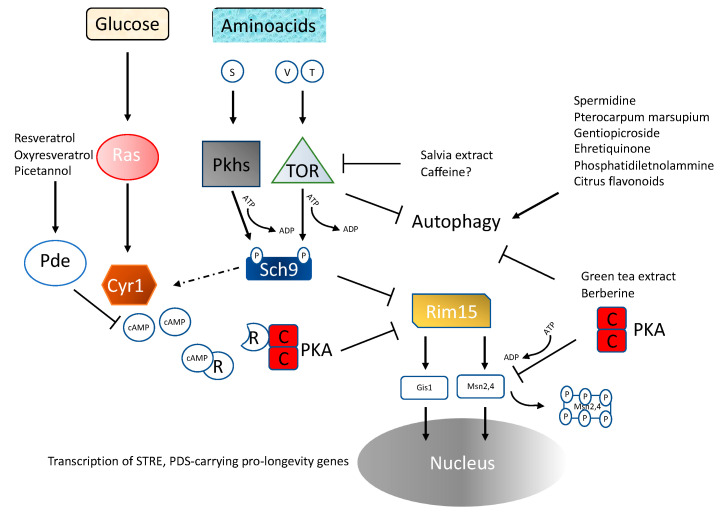
A detailed scheme of yeast longevity/nutrient pathways reporting the natural substances with modulating effects. Adapted from [34].

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
