# Peer review of "Yeast Chronological Lifespan: Longevity Regulatory Genes and Mechanisms"

_cells, 2022, doi:10.3390/cells11101714_

Round 1

Reviewer 1 Report

The manuscript of Mirisola and Longo provides a comprehensive and complete view of S.cerevisae as a central model to dissect out genes regulating age-dependent macromolecular damage and longevity extension. The review is well-written and well documented with up to 128 references among which historical references. However, a table recapitulating the most important genes and a scheme would helpful for the reader. The large amount of informations is indeed not easy to follow and illustrations may improve the readability for readers implicated in aging research. 

Author Response

Dear Reviewer, thank you for pointing at the need of figures and tables for a better understanding of the paper. We now added a table recapitulating the role of natural substances affecting yeast longevity and the related mechanism involved. In addition we added a figure of an experiment resuming the role of the major aging pathways and the role of fasting on yeast longevity. Finally an updated scheme of the gene network and related substances affecting longevity is provided.

Reviewer 2 Report

This manuscript aims to review the mechanisms associated chronological lifespan (CLS) . Started from different approaches in CLS studies, followed by  the mutation events in CLS, and ended with signaling pathways. Ras/PKA pathway, TOR/Sch9 pathway, shorten of amino acid or lipid, self-resource reuse, and oxidant status alteration are discussed in CLS. The manuscript provides collective information of CLS for the researchers. 

Concerns: 

  1. Some figures will help readers to understand it. For examples, the comparison of chronological lifespan and replicative life span. A figure of hallmarks in yeast CLS will also be appreciated.
  2. Side by side of yeast model and mammalian model comparisons will provide more impacts of the manuscript.
  3. There are some redundant parts in Sch9/TOR and Ras/PKA.

Author Response

(The authors gave the same response as above.)
